# A THD-Based Fault Protection Method Using MSOGI-FLL Grid Voltage Estimator

**DOI:** 10.3390/s23020980

**Published:** 2023-01-14

**Authors:** Wael Al Hanaineh, Jose Matas, Jorge El Mariachet, Peilin Xie, Mostafa Bakkar, Josep. M. Guerrero

**Affiliations:** 1Department of Electric Engineering, Polytechnic University of Catalonia (EEBE-UPC), 08019 Barcelona, Spain; 2Department of Energy Technology, Aalborg University, 9200 Aalborg, Denmark; 3Department of Electrical Engineering, Polytechnic University of Catalonia, 08222 Terrassa, Spain

**Keywords:** fault detection, fault protection, total harmonic distortion, distribution system

## Abstract

The rapid growth of the distributed generators (DGs) integration into the distribution systems (DSs) creates new technical issues; conventional relay settings need to be updated depending on the network topology and operational mode as fault protection a major challenge. This emphasizes the need for new fault protection methods to ensure secure protection and prevent undesirable tripping. Total harmonic distortion (THD) is an important indicator for assessing the quality of the grid. Here, a new protection system based on the THD of the grid voltages is proposed to address fault events in the electrical distribution network. The proposed protection system combines the THD with the estimates of the amplitude voltages and the zero-sequence component for defining an algorithm based on a finite state machine (FSM) for the detection, identification, and isolation of faults in the grid. The algorithm employs communication lines between all the protective devices (PDs) of the system to transmit tripping signals, allowing PDs to be coordinated. A second order generalized integrator (SOGI) and multiple SOGI (MSOGI) are used to obtain the THDs, estimated amplitude voltages, and zero-sequence component, which allows for fast detection with a low computational burden. The protection algorithm performance is evaluated through simulations in MATLAB/Simulink and a comparative study is developed between the proposed protection method and a differential relay (DR) protection system. The proposed method shows its capability to detect and isolate faults during different fault types with different fault resistances in different locations in the proposed network. In all the tested scenarios, the detection time of the faults has been between 7–10 ms. Moreover, this method gave the best solution as it has a higher accuracy and faster response than the conventional DR protection system.

## 1. Introduction

The increasing penetration of distributed energy resources (DERs), such as photovoltaic and wind energy systems, into the distribution systems (DSs) provides a great environmental and economic benefits, and also improves energy efficiency and reliability [1].

DSs are usually designed to be radial networks with unidirectional power flow. However, the deployment of microgrids (MGs), and DERs introduce the possibility of a bi-directional power flow. Furthermore, several technical challenges arise regarding this possibility, being the protection system a major issue, since the network should be protected from different kind of faults [2,3]. Fault protection in DSs is a challenging and complex issue, especially considering distributed generation (DG), as the power flow changes depending on the network topology and operational mode. Therefore, the short circuit fault current value will be affected in some cases, which might lead to a miscoordination of the protective devices PDs [4,5].

Moreover, in MGs, the utility grid is the source of the majority of faults during the grid-connected mode of operation, which results in very large fault currents. However, for islanding mode operation, the power limitations of the semiconductor devices mean that the faulted currents are much smaller and could be not enough to trigger a breaker [6,7]. Under such circumstances, some traditional protection systems are unable to operate properly and lead to protection blinding and false tripping. Therefore, detecting the fault accurately can significantly increase the network performance [8,9,10].

In recent years, considerable efforts had been made to develop different techniques that can locate and detect faults in DSs despite the presence of multiple fault sources [3]. These techniques can be grouped into: conventional protection technologies that use three types of relays as a main protection component: directional over current, distance, and differential relays [11,12,13,14,15,16]. And voltage-based protection [17,18], adaptive protection [19,20], and harmonics-based protection methods [21]. Each of these types have their own advantages and weak points. Table 1 shows a comprehensive comparison of these techniques.

The integration of DGs offers different paths to deliver the electric power to the customers. The fault protection process is complicated, and the introduced possibility of bi-directional power flow imposes a major challenge in protection schemes, which threatens the mentioned methods. This emphasizes the need for new protection methods to ensure the system safety regardless of the direction of the power flow. Moreover, one of the main reasons to develop a new protection method is to reduce the time required by maintenance crews to restore service in case of power outage, which increases the network efficiency. A fast restoration can be achieved by a fast and accurate detection and location of the fault. Therefore, the approach proposed here can provide fast fault protection process with a reasonable implementation cost, unlike conventional methods.

In this paper, a new protection method based on the novel THD measurement method reported in [22] is proposed for solving the previous problems. In [22], a SOGI filter with few additional math operations was proposed for obtaining the THD of a signal, which was implemented into a digital signal processor (DSP) of Texas Instruments with a low computational burden. In this case, the method is applied with a MSOGI approach to perform the detection process of faults in the grid. The THD is computed in the MSOGI considering only the fundamental and the triple-*n* harmonics of the grid voltages: the 3rd, 6th, and 9th harmonics [23], for simplicity of implementation. The MSOGI also provides the estimated amplitude voltages and the zero-sequence components of the grid with a small extra computational cost. The protection algorithm is performed using an FSM algorithm that is in charge of triggering the detection, identification, permanence checking, and isolation of the fault [24,25].

The proposed method provides a fast detection and isolation for symmetrical and unsymmetrical faults with a computational load that could be implemented into a DSP, or in a field programmable array (FPGA), since the MSOGI is much more affordable to implement than an equivalent FFT version. An assessment of affordability is provided at the end of the paper. To the Authors’ knowledge, this is the first time that such proposal has been presented. The previous methods required the use of an FFT. The proposed method has been examined while considering different fault types, fault locations in the proposed network, and fault resistances. It has been also compared with a conventional differential relay protection method under similar conditions.

The rest of the paper is structured as follows: Section 2 presents the proposed protection algorithm and the FSM in detail. Section 3 presents the THD method using the MSOGI and the estimated amplitude voltages and the zero-sequence components. Simulations and comparison with a DR system are carried out to validate the performance of the proposed method in Section 4. Finally, the conclusion is provided in Section 5.

## 2. Protection Algorithm

A protection algorithm is defined so that each PD is able to detect the fault location. The algorithm is based on the THD levels, the estimated amplitude voltages, and the zero-sequence components of the three-phase voltage signals of each PD. The zero-sequence components are used to identify phase-to-phase faults. An FSM that employs six states in charge of operating through the protection process is defined so that each PDs is able to locate, check the permanence, and isolate the fault. The scheme of the proposed method is shown in Figure 1.

### 2.1. Proposed System

The protection method can be applied to different DSs. Figure 2 shows a single line diagram of the proposed DS with DG penetration used for testing the method. The system is composed by a 33 kV grid with a main-line (L3), PD5, and breakers (CB9 to CB11) passing through a step-down transformer connected to two distribution lines (DLs), (DL1 and 2). The parameters of the system are listed in Table 2. Each DL has a PD and breakers (PD3, CB5 and 7) at DL1 and (PD4 CB6 and 8) for DL2 and connected to a MV/LV transformer. Two distributed generators (DG1 and 2) and local loads (Load1 and 2) are connected at the end of the DLs and placed in two separated zones (Zone1 and 2). Each zone has its own PD and breakers, (PD1, CB1 and 3) for Zone1, and (PD2, CB2 and 4) for Zone2. Moreover, communication links between the PDs are defined and intended for transmission of tripping signals for the coordination and isolation of the faulted locations.

Faults can happen in different locations of the DS. In this work, the faults are studied in three different scenarios. Note that the fault type of each scenario is shown in Figure 2 and explained in Section 4.

Scenario I. A fault defined as F3 occurs at L3. Then, PD5 detects and isolates the fault, which gives the chance for the DGs to supply the loads by its own.Scenario II. A fault labeled as F2 happens at DL2. PD4 is in charge of detecting and isolating the fault, while DL1 remains connected to the system, supplying Load1.Scenario III. A fault defined as F1 occurs inside Zone1. In this case, PD1 detects and isolates the fault, which allows the rest of the system to remain connected.

### 2.2. Fault Classification Algorithm

Faults might occur in one or more phases of the grid to the ground or between the phases only. In this work, a pre-processing stage is implemented to measure the three-phase voltages, calculate the THD, defined as $THD_{abc}$, the estimated amplitude voltages, defined as ${\tilde{A}}_{abc}$, and the zero-sequence components, defined as $V_{abc0}$, for identifying the fault-type and be able to perform a decision by the FSM. The fault detection is made using two thresholds: one named *α* for comparing with $THD_{abc}$; and the other named Δv for comparing with ${\tilde{A}}_{abc}$. *α* is set to 5%, which is the recommended IEEE standard 519–2014 value of the voltage harmonic distortion for DSs [26]. Δv is set to 7.5%, which is the acceptable grid voltage drop range at DSs voltage level according to the technical requirements in the Spanish grid code for reliable energy integration [27].

At a fault event, an abrupt increase in the measured $THD_{abc}$ is produced at the same time as a sudden drop in ${\tilde{A}}_{abc}$, making their values be out of the defined thresholds ranges. Then, at this point, the zero-sequence voltage is used to distinguish between phase-to-phase (2PH) and phase-to-phase-ground (2PH-G) faults, as both have the same conditions at the beginning of the fault. As there are no zero-sequence components during 2PH faults [28], then if $V_{abc0}=0$ the fault is 2PH and 2PH-G if $V_{abc0}\;\neq \;0$. The zero-sequence voltages are calculated as follows:(1)$$V_{abc0}\;=\frac{1}{3}\left(v_{a0}+v_{b0}+v_{c0}\right),$$

These values are used to identify the fault and adopt a decision. The faults had been classified to belong to eleven categories numbered from 0 to 10, as seen inside Figure 3.

### 2.3. Finite State Machine

In Figure 2, a total of five PDs are defined; each PD is in charge of an FSM to locate and isolate the faults in different locations of the DS. Fault isolation implies disconnection of the faulted DL while keeping the others connected and supplying their corresponding zones. The FSMs are also designed to detect permanent and temporary faults. A temporary fault is a spurious fault with a duration of less than 100 ms that disappears without need for any protection action [25]. The behavior of the FSMs is tested with the defined fault scenarios, F1–F3, see Figure 2.

The time in which the fault is detected by the FSM of each PD is different, which indicates that locating and isolating a fault can be achieved. The time depends on the fault location. the FSM of the PD nearest to the faulted line detects the fault more quickly than the other FSMs and will use the communication links to send a fault-message to the other FSMs more quickly, allowing their process to be stopped and give the concerned FSM an opportunity to isolate the fault. For example, for a fault at F3, FSM5 will detect the fault more quickly than the other FSMs, whereas for a fault at F2, FSM4 will detect the fault more quickly. These can be seen later in the figures of Section 4.1.1 and Section 4.1.2, respectively.

The time in which the message arrives to every FSM depends on the communication technology and might be affected by specific communication delays [29], defined here as $T_{d}$. The protection system can employ several methods to communicate between PDs. In this work, optical fiber technology is chosen as the communication link between the PDs of the network. Optical fiber is characterized by its fast transmission speed, security, and low latency, which are able to cover long distances [29]. The signals exchange in our system is expected to be based on the IEC 61850 standard with an estimated delay of 10 ms [30].

The communication delay allows the protection system to be coordinated and to operate normally. However, in this work, it is supposed that an extra delay, between 10 ms and 100 ms, can be produced. In this case, the FSM of a particular PD will receive the message in the middle of its process, allowing it to stop and to wait for the fault to be removed and to return to normal operation. However, if the delay exceeds 100 ms, the FSM makes its own decision which could lead to a CB being tripped.

The FSMs are defined using six states that are described below and depicted in Figure 4, to simplify the explanation.

State S1. Normal Operation:

The FSM enters this state initially and remains in it; meanwhile, the system operates inside a boundary of 1 ± Δv pu around nominal values and, at the same time, $THD_{abc}<\alpha$. Therefore, the FSM is waiting for a fault.

State S2. Fault Detection

The FSM enters this state when a fault is detected, i.e., $THD_{abc}>\alpha$ and ${\tilde{A}}_{abc}<\left(1-\Delta v\right)$ pu, which means that any of the phases reach these two conditions. A timer, named as $T_{c}$, is started inside the FSM that counts up to measure the fault duration. A fault signal is sent to the other FSMs by means of the communication links to advise them of the event.

The behavior of the THD measurement method produces a peak in the THD, at the time that the fault happens. This peak decays exponentially with time and drops to zero. So, when it reaches $THD_{abc}<\alpha$ while ${\tilde{A}}_{abc}<\left(1-\Delta v\right)$ pu still holds the FSM changes to S3. Notice that the peak behavior in the THD and exponential decay can be seen in the figures of Section 4.1.1 and Section 4.1.2, respectively.

Meanwhile in S2, if the fault disappears, as in case of a temporary fault, then $THD_{abc}<\alpha$ and ${\tilde{A}}_{abc}$ returns to nominal values and the FSM returns to S1 (normal operation). At the same time, if a fault message is received from the other FSMs, the FSM goes to state S6, which is intended to hold the non-faulted FSMs.

State S3. Waiting for the Fault to be Permanent

During S3, the timer $T_{c}$ continues counting. If $T_{c}$ reaches the maximum count-up of 100 ms then the FSM goes to state S4, intended for fault isolation. Meanwhile if $T_{c}<100\;\mathrm{ms}$ and the fault disappears (${\tilde{A}}_{abc}$ returns to nominal) during S3, then the FSM goes back to S1 (normal operation) and no tripping action is taken.

As in the previous state, during S3 and $T_{c}<100\;\mathrm{ms}$, if the FSM receives a fault message, it goes to state S6. This means that the fault is not inside the location of the FSM.

State S4. Fault Isolation

The FSM enters this state only when $T_{c}$ reaches 100 ms, meaning that the fault is permanent. This breaks the algorithm and sends an active trip signal to the nearest CB, isolating the fault. Furthermore, arriving at S4 in the non-faulted FSMs implies that these FSMs have not received a fault message from the faulted FSM. After the tripping, $T_{c}$ is reset to 0 and the FSM goes to S5, waiting for actions to be made for a grid reconnection.

State S5. Grid Reconnection

This state is defined as awaiting actions from the grid operator or maintenance personnel to re-connect the line and return the FSM to S1.

State S6. Holding non faulted FSMs

The FSM enters this state when it receives a fault signal from a faulted FSM during S1, S2, or S3. At S6, the FSM waits for the fault to disappear or be isolated. The waiting action is performed by starting a count-up on another timer named as $T_{w}$. When $T_{w}$ reaches 100 ms, the FSM returns to S1.

## 3. MSOGI-FLL THD Measurement Method

The THD measurement uses the square root of the sum of the squared harmonic components of a given signal, divided by the fundamental component, which is derived from the standard definition [31,32] given by:(2)$$THD=\frac{\sqrt{\sum_{h}{\left|A_{h}\right|}^{2}}}{A_{1}}\cdot 100,$$
where *h* and $A_{h}$ are the harmonic order and the amplitude of the *h*^th^-harmonic component, respectively, for h ≠ 1, and $A_{1}$ is the amplitude of the fundamental component. The measurement in this paper uses the MSOGI [33] for providing the harmonics needed for the THD calculation. The MSOGI is based on multiple SOGIs operating simultaneously in parallel to obtain the estimate of the fundamental and harmonics components of the grid (see Figure 5). In this figure, $v_{in}$ is the input voltage signal, e is the error, and $\tilde{\omega }$ is the estimated angular frequency. The MSOGI uses a cross-feedback cancelation network that removes the unnecessary components from the input signal in a way that every SOGI receives only the component that is intended to extract. A FLL is used to give the MSOGI (MSOGI-FLL) the capability to follow the grid operative frequency. The FLL is linked with the first SOGI, which provides the fundamental component to this block. It is important for the FLL to deliver $\tilde{\omega }$ to all the paralleled SOGIs. In this work, the MSOGI will also be affected by faults that will distort the estimated $\tilde{\omega }$, which will further distort all the system. Then, a saturation block is used at the output of the FLL to limit the distortion range on $\tilde{\omega }$ to not more than ± 1Hz around the grid’s nominal frequency.

For the MSOGI-FLL, the parameter *ξ* is set to $1/\sqrt{2}$ to achieve an optimal relationship between transient response speed and rejection to harmonic distortion [33]. Only multiples of the third harmonic are used to calculate the THD for not having to be distorted by other harmonics in the grid. Only the triple-*n* harmonics, the 3rd, 6th, and 9th, are used in the MSOGI-FLL. These triple-*n* harmonics have the special characteristic of only being contained in the power inverter’s AC-side neutral point [23]. Therefore, these harmonics will not be present in the grid lines during normal operation and cannot affect the MSOGI-FLL. Moreover, when the other harmonics appear during normal grid functioning, they could not affect the THD since they are not included in the MSOGI scheme. However, in the event of a fault, a sharp fall in the voltage phases is produced that excites almost all the harmonic components but has no relationship with the mentioned triple-*n* harmonics contained in the neutral point, so the THD will be particularly affected. Therefore, this concept indicates that the obtained THD is independent of the previous harmonic distortion of the grid during normal operation. It is important to note that the THD proposal used here is an instantaneous indication that a fault has been produced, and in this case it functions as a kind of fault sensor for the grid. This THD does not represent the real THD of the grid voltage, which has to consider all the harmonic components, as was in fact the objective of the first proposal of [22]. This THD allows the fault to be detected quickly, which is used as the kernel of this protection algorithm.

The MSOGI-FLL is used at each phase of the grid voltage. The fundamental and 3rd, 6th, and 9th amplitude of the harmonic components are obtained using the in-phase and quadrature-phase outputs of each SOGI as:(3)$${\tilde{A}}_{h}\;\approx \;A_{h}=\sqrt{v_{dh}^{2}+v_{qh}^{2}},$$
where h is the index of the component. Then, the THD is obtained by using the definition in (2), which involves few math operations: sum of the squared harmonic components, a square root, and a division, as shown in Figure 6. The obtained THD is denoted as ${\tilde{v}}_{THD}$. Notice that multiplication by 100 is only necessary to achieve a percentage scale and that saturation is used to avoid a possible division by zero at the starting of the system. A second order LPF is designed in order to obtain a desired balance between speed transient response and distortion levels in the obtained THD signal ${\tilde{v}}_{THD}$.

Regarding the zero-sequence component, Figure 7 depicts the used calculation scheme, following (1). Notice that only the in-phase voltages of the first SOGI of the MSOGI shown in Figure 5 are used in each phase of the grid voltage, i.e., $v_{a0}$, *v*_*b*0_, and $v_{c0}$ are used in the calculation.

## 4. Results and Discussion

Different case test scenarios had been performed using MATLAB/Simulink software to validate the proposed protection method. Simulations had been carried out taking into account different conditions as the change of fault types, fault resistance, and fault location for the network as depicted in Figure 2 and with the parameters shown in Table 2.

### 4.1. DLs Protection Test

In this section, two types of fault are examined in F2 and F3 locations, at 0.2 s (see Figure 2).

#### 4.1.1. Three-Phase Fault (3PH-G) at F3

Figure 8 depicts $THD_{abc}$, ${\tilde{A}}_{abc}$ and FSM states when a symmetrical (3PH-G) fault occurs at F3. Notice in Figure 8 how $THD_{abc}$ increases abruptly at 0.2 s. The fault is detected when $THD_{abc}>5\%$ and then FSM5 goes from S1 to S2. At the same time, the low fault resistance (*r* = 0.001 Ω) causes ${\tilde{A}}_{abc}$ to drop towards 0 pu. The fault is identified when ${\tilde{A}}_{abc}<0.925$ pu. At this moment, FSM5 sends a fault message to the other non-faulted FSMs and starts a count-up on its own $T_{c}$ timer. When the other FSMs receive the message, they move from state S1 to S6 and start their own count-up $T_{w}$ timers, waiting for the fault to be isolated. The $THD_{abc}$’s behavior of FSM5 suffers from a peak that decays exponentially to zero after a short time. Then, when $THD_{abc}$ returns to $THD_{abc}<5$% while ${\tilde{A}}_{abc}<0.925$ pu still holds, FSM5 goes from S2 to S3, and $T_{c}$ continues counting up. Whenever $T_{c}$ reaches $T_{c}=100\mathrm{ms}$, the fault is declared permanent and FSM5 moves from S3 to S4, triggering a signal that breaks the algorithm and trips CB10 and CB11 to isolate the fault and disconnects L3 from the grid. Additionally, $T_{c}$ is reset to 0 and FSM5 goes from S4 to S5, waiting actions to re-connect L3 to the grid. Notice that in FSM5, in the transition from S3 to S4, a momentary disruption in the grid happens due to L3 disconnection, which causes to spike again. However, this does not suppose something since the algorithm is in state S4 and does not care about the THD. Moreover, CB9 is triggered to isolate reversed flow in DGs and to allow the power to be shared between the loads. At the non-faulted FSMs, whenever $T_{w}$ reaches 100 ms, these FSMs return to S1 (normal operation).

#### 4.1.2. Phase-to-Phase Fault (2PH) at F2

In this case, a long delay in the communication between PDs is considered. Figure 9 depicts $THD_{abc}$, ${\tilde{A}}_{abc}$ and FSMs states during an unsymmetrical 2PH fault at F2. A fault between phases *b* and *c*, BC-fault, is considered at 0.2 s that makes $THD_{bc}$ increase abruptly. The fault is detected by FSM4 when $THD_{bc}>5\%$ while $THD_{a}<5\%$. In this case, FSM4 goes from S1 to S2 (Fault detection). Meanwhile, the absence of the ground connection and to the fault resistance (r=0.1 Ω) causes ${\tilde{A}}_{bc}$ to drops towards 0.5 pu, and ${\tilde{A}}_{a}$ remains unaffected (1 pu). The fault is then identified when ${\tilde{A}}_{bc\;}<0.925$ pu and $V_{abc0}$ are shown to be zero. Then, the $T_{c}$ internal timer of FSM4 starts counting up and a fault message is sent to the non-faulted FSMs. In this fault case scenario, a delay of 30 ms has been intentionally added to the PDs communication links to show its effect on the protection system. Therefore, prior to receiving the message, the non-faulted FSMs detects the fault and also moves to S2. Notice in Figure 9 that one of the non-faulted FSMs detects the fault (i.e., moves to S2) at time 0.223 s, i.e., PD3 goes to S2 23 ms later because of how the distance between the location of the fault and this PD causes it to see the fault. Note that all non-faulted FSMs (i.e., FSM 5, 3, 2, and 1) are assumed to see fault at the same time (after 23 ms), for simplicity of explanation. However, at the moment in which the FSMs receive the message (at 0.2375 s and 30 ms later because of the delay), they move from S2 to S6 (Hold state), and $T_{w}$ starts counting up. As in the previous fault case, after a short time, the $THD_{bc}$ signals decay exponentially to zero. Therefore, when they reach $THD_{bc}<5\%$ while ${\tilde{A}}_{bc}<0.925$ pu, FSM4 goes from S2 to S3 (Waiting the fault to be permanent), and $T_{c}$ continues counting up. Whenever $T_{c}$ reaches 100 ms, the fault is declared to be permanent. FSM4 moves to S4 (Fault isolation) and sends an active trip signal to CB6 and CB8 to isolate the fault. Then, $T_{c}$ is reset and FSM4 goes to S5, waiting for reconnection actions. At this moment, in the non-faulted FSMs, the time $T_{w}$ continues to count up, and whenever $T_{w}$ reaches 100 ms they return to S1 (Normal operation). Note that Zone2 operates using its own DG, and that during S4 (Fault isolation), all THD signals spike due to DL2 disconnection, which supposes an abrupt change in the grid voltages, but this does not affect the protection algorithm since it is in S4.

Figure 10a show the grid currents ($i_{abc}$) for the 3PH fault at L3 and Figure 10b for the 2PH fault at DL2. Note in Figure 10a that only PD5 trips at 0.307 s, while the other PDs do not trip. In Figure 10b note that only PD4 trips at 0.3075 s.

### 4.2. DGs Zones Protection Test

The protection has been examined in this fault case scenario at the DGs zones in the case of having a 5th harmonic distortion in the grid when a 2PH-G fault occurs at F1 in Zone1.

Figure 11 shows the behavior for a fault between phases *b* and *c* to ground which shows that the 5th harmonic distortion does not affect $THD_{abc}$ before the fault, since this harmonic is not considered in the MSOGI. In a 2PH-G at 0.2 s, $THD_{bc}$ abruptly increases, so $THD_{bc}>5\%$ and $THD_{a}$ remain at 0%. Then, FSM1 detects the fault and goes to S2 (Fault detection). At the same time, ${\tilde{A}}_{bc}$ drops towards 0 pu due to fault resistance (r=0.001 Ω), ${\tilde{A}}_{a}$ remains unaffected. Then, when ${\tilde{A}}_{bc}<0.925\;\mathrm{pu}$ and $v_{abc0}\;\neq \;0$ is checked and the fault is identified, FSM1 sends a fault message to the other non-faulted FSMs and starts its own count-up $T_{c}$ timer. These FSMs receive the message (at 0.218 s, which is due to the normal delay of the communication links) and move from S1 to S6 (Hold state), and $T_{w}$ starts counting up.

At FSM1, when $THD_{bc}<5\%$ while ${\tilde{A}}_{bc}<0.925$ pu, it goes from S2 to S3 (Waiting for the fault to be permanent), meanwhile $T_{c}$ continues counting up. Whenever $T_{c}$ reaches 100 ms, the fault is considered permanent, and FSM1 moves to S4 (Fault isolation) and sends an active trip signal to CB1 and CB3 to disconnect DG1 and isolate Zone 1. Then, $T_{c}$ is reset and FSM1 goes to S5, waiting for reconnection actions. At this moment, in the non-faulted FSMs, the time $T_{w}$ continues to count up, and whenever $T_{w}$ reaches 100 ms they return to S1 (Normal operation). Notice that during S4 (Fault isolation), the disconnection at time 0.308 s causes all the THD signals to spike, but this does not affect the protection algorithm since it is in S4.

Figure 12 shows the grid currents for a 2PH-G fault at Zone 1. In this case, only PD1 trips at 0.308 s as it is the nearest PD in detecting the fault.

These simulation results show that the proposed method has the capability to detect, identify, and isolate the fault in the least possible time in different locations and scenarios. The time from a fault event to the detection of the fault by the proposed method has been measured. For the 3PH, the detection time is 7 ms. For the 2PH (BC) fault, the detection time is 7.5 ms. Finally, for the 2PH-G (BC-G) fault, the detection time is 8 ms. Note that in all the cases, the involved time is less than 10 ms.

### 4.3. Comparison with Different Protection Methods

The Differential Relay (DR) protection method explained in [34] is tested using the proposed system in Figure 2 to validate and emphasize the proposed protection method advantages. The DR protection can overcome the problem of bidirectional power flow in a DS [3,35]. However, if the current transformers are saturated or incorrectly configured, a DR may have tripping issues [36]. Additionally, the DR relay parameters must be adjusted to the different grid’s changing circumstances [37].

In this section, a 2PH (BC) fault at F2 with a fault resistance of 0.1 Ω is considered at 0.2 s to compare the DR with the proposed method exposed in Section 4.1.2. The DR relay settings are shown in Table 3.

The performance of the DR protection method is shown in Figure 13. The DR had been designed to take an action after 100 ms of the fault detection for simplicity of comparison. Note that the fault had been cleared at time 0.313 s, which, considering the 100 ms, means that the DR takes 13 ms to detect the fault. This is slower than the THD proposed method, which clears the fault at time 0.3075 s, meaning that it takes 7.5 ms, as shown in Figure 10b. The DR takes more time because it depends on the differential current value.

Furthermore, one of the difficulties in distribution network fault diagnosis is the detection of different fault impedances, so in this regard the two protection methods had been tested considering different fault resistances, fault types, and fault locations. For example, when a 1PH-G (AG) fault with a fault resistance of (r=5 Ω) occurs in L3 at 0.2 s, it has been noticed that the MSOGI-THD method clears the fault at 0.3095 s while the DR method clears the fault at 0.3152 s. Figure 14 shows the grid currents for a 1PH-G fault at L3 for both methods. In all cases, the proposed method clears the fault faster than the DR method, as shown in Figure 15. Moreover, the proposed method is compared with other protection methods under the same conditions. Table 4 summarizes the tripping times and shows the advantages and disadvantages of the different methods. It can be stated that the proposed THD protection method can be a viable solution to be adopted for providing a faster trip decision under various conditions when using communication lines.

### 4.4. Simplification of the MSOGI Structure

The method proposed here uses a MSOGI structure for each phase of the grid. Each MSOGI employs four SOGIs: one for the fundamental and three more for the 3rd, 6th, and 9th harmonics, respectively (see Figure 5). This is the biggest computational part of the algorithm. Therefore, to simplify the algorithm and try to minimize the computational burden, the number of harmonics employed in the MSOGI can be reduced, using fewer SOGIs in their inner structure, without affecting the THD calculation and the protection system operation.

According to [22], a SOGI-FLL was implemented in a Texas Instruments (TI) Concerto F28M35H52C1 DSP control board using Code Composer Studio software environment from TI [44]. The Concerto DSP is a dual core processor that has an ARM Cortex-M3 and a TMS320C28x inside the same chip. The SOGI-FLL was implemented in the TMS320C28x, which is a 32-bit floating point processor that runs at 150 MHz clock speed and has 512 kb Flash memory. The computational burden of the SOGI-FLL was computed in number of processor cycles (c) used to compute the SOGI and the FLL blocks. In [22] it was reported that the SOGI needed 149c and the FLL 49c.

The MSOGI has been simplified using only the 6th and 9th harmonics, which means that it needs three SOGIs per phase and a total of nine SOGIs per method. This approach is named here as “MSOGI-II”. Minimal implementation had been also adopted considering only the 9th harmonics, which means that two SOGIs per phase and six SOGIs per method were needed, named here as “MSOGI-III”. Moreover, the number of adders in the cross-feedback network is reduced to nine per phase for the MSOGI-II and to six per phase for the MSOGI-III.

A 3PH-G fault at F3 with a fault resistance of 0.001 Ω is considered at 0.2 s to demonstrate the comparison of the MSOGI-II and MSOGI-III with the MSOGI approach exposed in Section 4.1.1. The fault clearing times of the three approaches are given in Table 4. The results indicated that the three approaches’ fault-clearing times are quite close, which validates the fact that the algorithm can be simplified without significantly compromising its capability for protecting the system. Finally, the MSOGI-III approach can be used for the protection system. Note that Table 5 also shows the number of processor cycles required for executing the SOGI part of each MSOGI approach.

In [22], an FFT also was implemented into a DSP and the computational burden given in processor cycles. The FFT was computed using the subroutines of the optimized library provided by TI [45]. The FFT was reported to need 7019 cycles per phase, which means that using an FFT to compute the THD of the three-phase system will require around 21,057 of processor cycles, which clearly justifies the MSOGI-THD proposal.

## 5. Conclusions

This paper has proposed an algorithm based on FSMs to detect, identify, and isolate faults in different locations of a DS grid. The algorithm used the THD levels of the grid voltages for fault detection, then the estimates of the amplitude voltages and the zero-sequence components are used for fault identification. An FSM has been defined for each PD, which employs six states in charge of operating through the protection process in order to locate and isolate the faults, as well as to detect permanent and temporary faults.

The MSOGI-FLL has been used to extract the triple-*n* harmonics, i.e., the 3rd, 6th, and 9th harmonics, that have been used as the kernel of the algorithm to obtain the THD. These harmonics are usually only contained in the neutral points of the power system, but are also excited transitorily at a fault event, which have been allowed to perform a fast detection of the fault and do not have conflict with the rest of the grid harmonics that might be present in the grid before the fault. The employed MSOGI-FLL THD measurement method supposes a computational burden much more affordable to employ for a digital processor than a standard FFT, as it is obtained by using a few math operations, namely: sum of the squared harmonic components, square root, and division.

This new approach can be used as a fault sensor for the electrical network, and this work shows that it can be used to improve the system’s accuracy and speed up the response of the protection system. The simulation results of Section 4 showed that the algorithm has the capability to detect, identify, and isolate different types of faults in different locations and under different scenarios. The effectiveness of the proposed method was shown through a comparison with a conventional DR and with other protection methods. In all the studied cases, the detection time of the faults have been measured to be between 7–10 ms. The proposed protection method demonstrated its practical benefits over other conventional protections methods in terms of trip time response speed of the PD when compared with other protection methods operating in similar conditions.

Furthermore, the structure of the MSOGI had been simplified and minimized. The obtained results demonstrate that the minimal expression of the MSOGI (the MSOGI-III) could be used in the protection scheme without compromising the effectivity of the proposed algorithm.

In the future, the method could be applied to systems with more complex distribution lines, such as ring distributed systems and with different high voltage levels, and such applications could be investigated. Moreover, research can be performed to try to avoid communication use in the system.

## Figures and Tables

**Figure 1 sensors-23-00980-f001:**
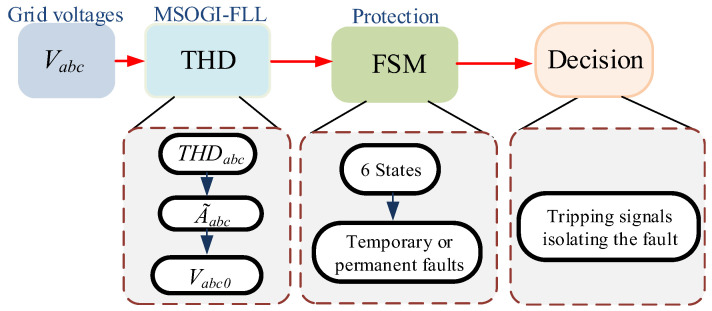
Scheme of the proposed method.

**Figure 2 sensors-23-00980-f002:**
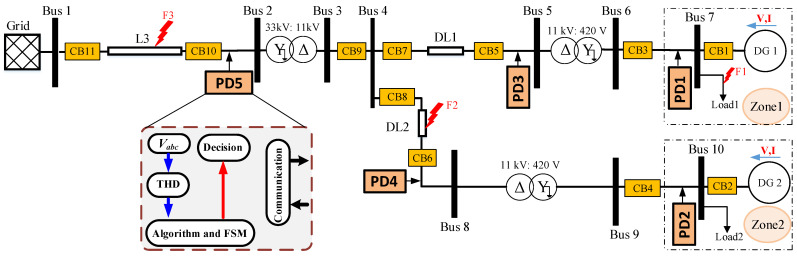
Single line diagram of the proposed system.

**Figure 3 sensors-23-00980-f003:**
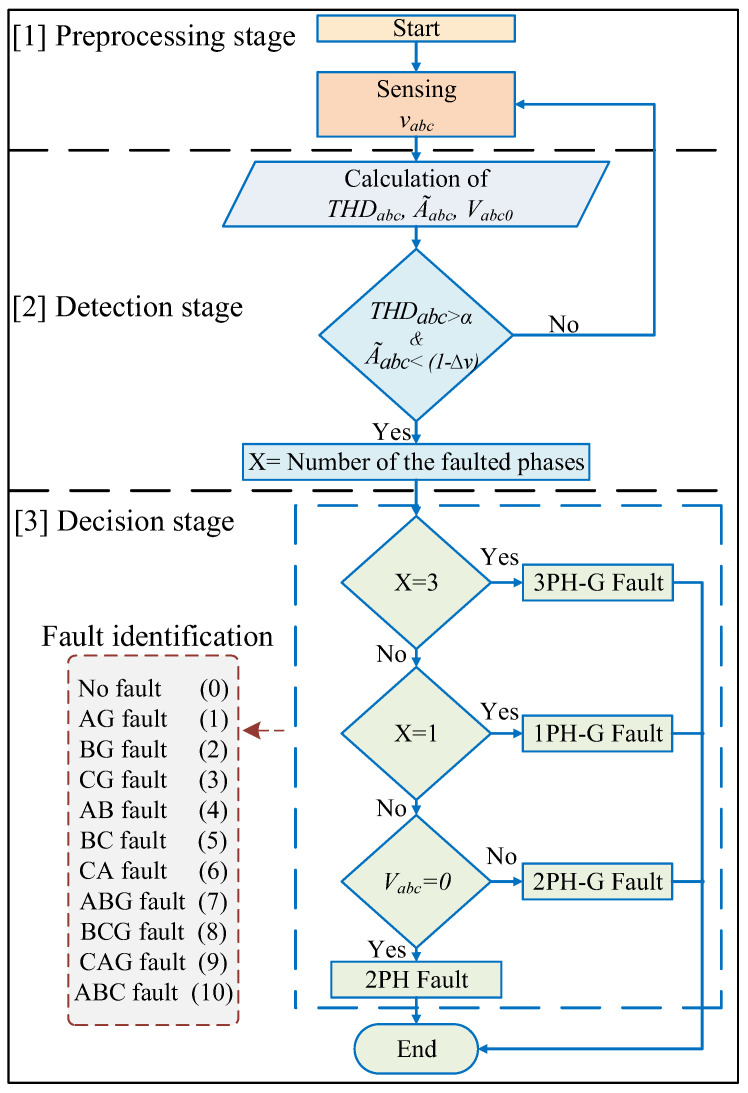
Flowchart of the fault classification algorithm.

**Figure 4 sensors-23-00980-f004:**
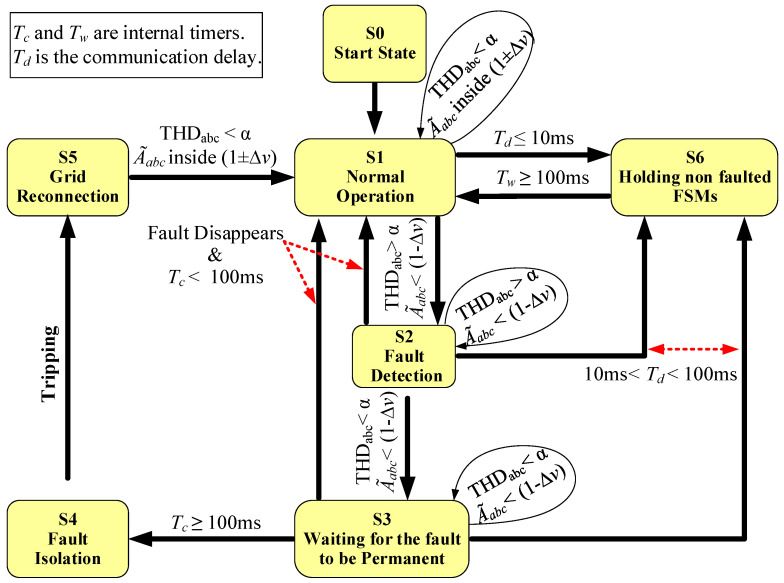
Scheme of the proposed FSM.

**Figure 5 sensors-23-00980-f005:**
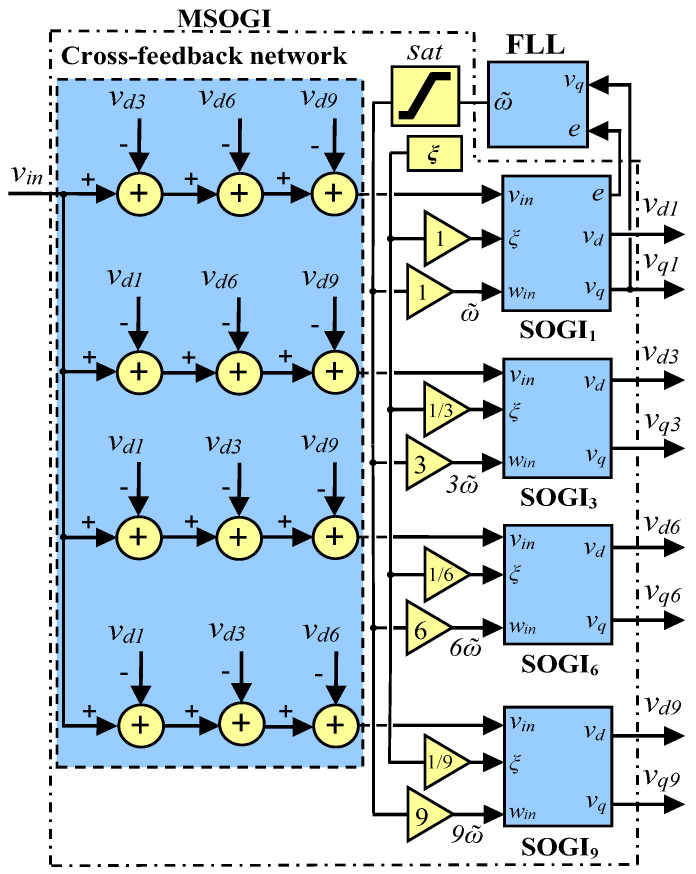
Block diagram of the MSOGI-FLL.

**Figure 6 sensors-23-00980-f006:**
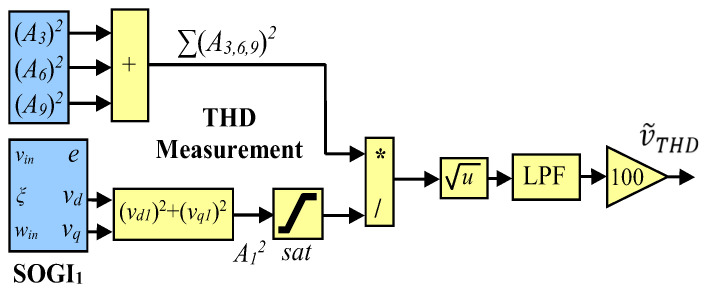
Block diagram of the MSOGI-FLL THD method.

**Figure 7 sensors-23-00980-f007:**
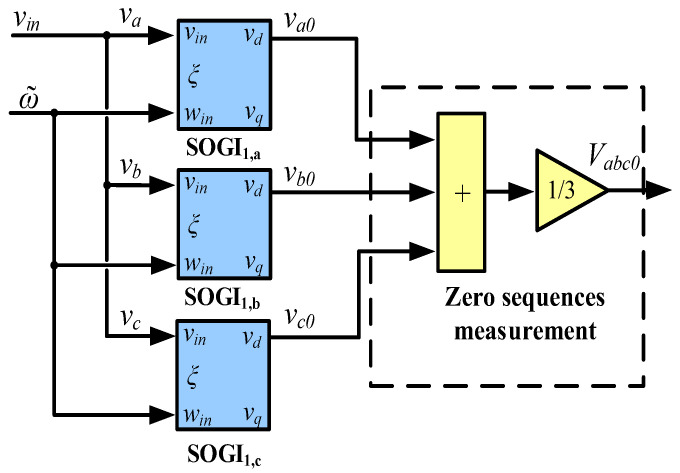
$V_{abc0}$ zero-sequence detection block-scheme.

**Figure 8 sensors-23-00980-f008:**
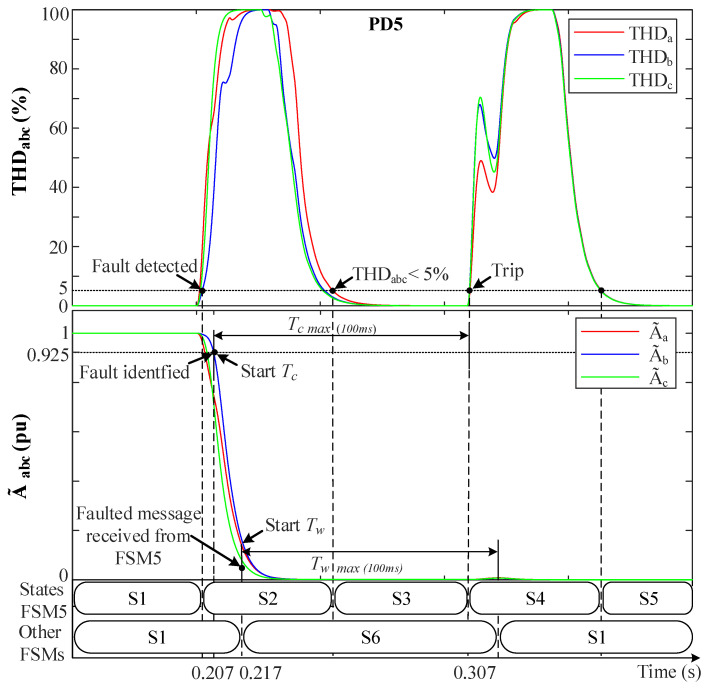
The system behavior with the FSMs during a 3PH-G fault at F3. (**Upper**) $THD_{abc}$. (**Lower**) ${\tilde{A}}_{abc}$.

**Figure 9 sensors-23-00980-f009:**
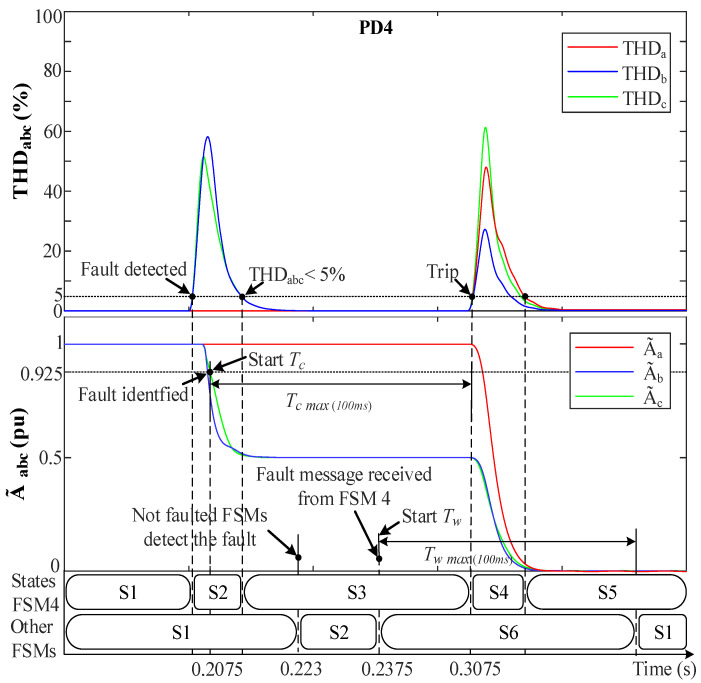
The system behavior with the FSMs during a 2PH fault at F2. (**Upper**) $THD_{abc}$. (**Lower**) ${\tilde{A}}_{abc}$.

**Figure 10 sensors-23-00980-f010:**
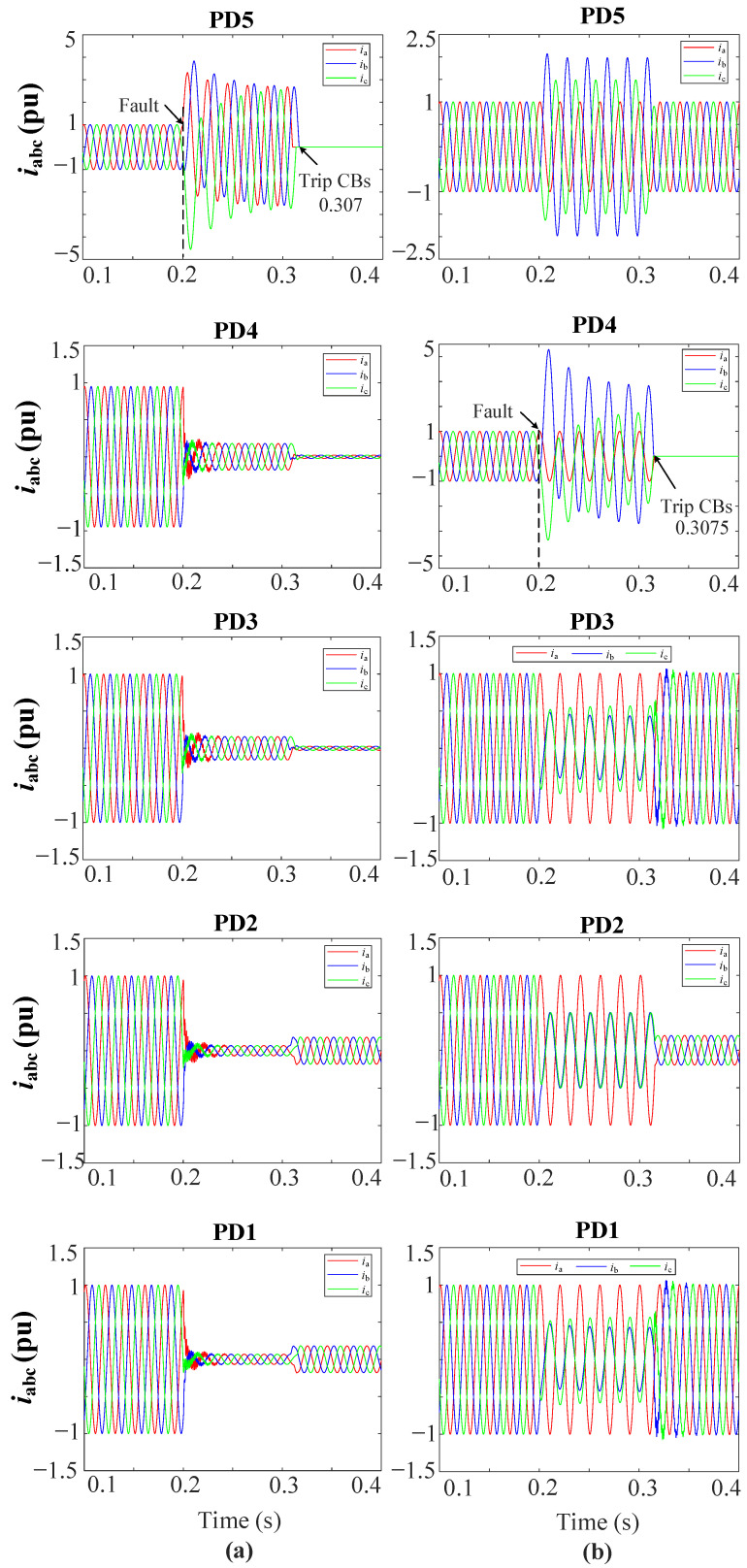
$i_{abc}$ currents for the grid (**a**) during a 3PH fault at L3, (**b**) during a 2PH fault, phases *b* and *c*, at DL2.

**Figure 11 sensors-23-00980-f011:**
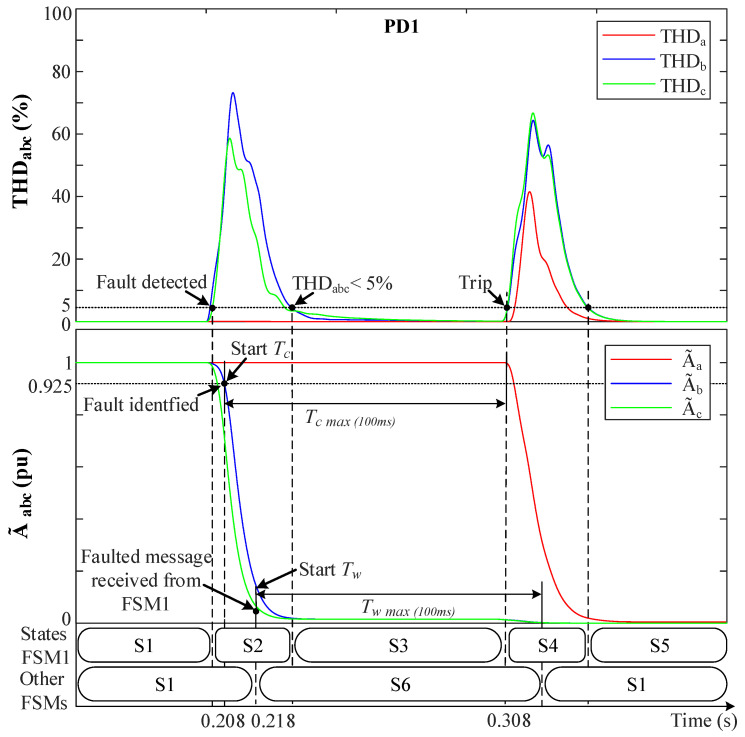
The system behavior with the FSMs during a 2PH-G fault at F1. (**Upper**) $THD_{abc}$. (**Lower**) ${\tilde{A}}_{abc}$.

**Figure 12 sensors-23-00980-f012:**
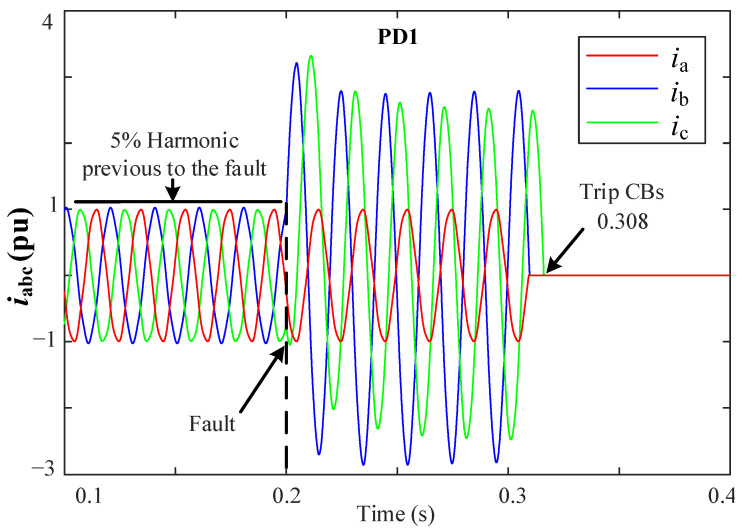
$i_{abc}$ currents during a 2PH-G fault at Zone1, phases *b* and *c* to ground.

**Figure 13 sensors-23-00980-f013:**
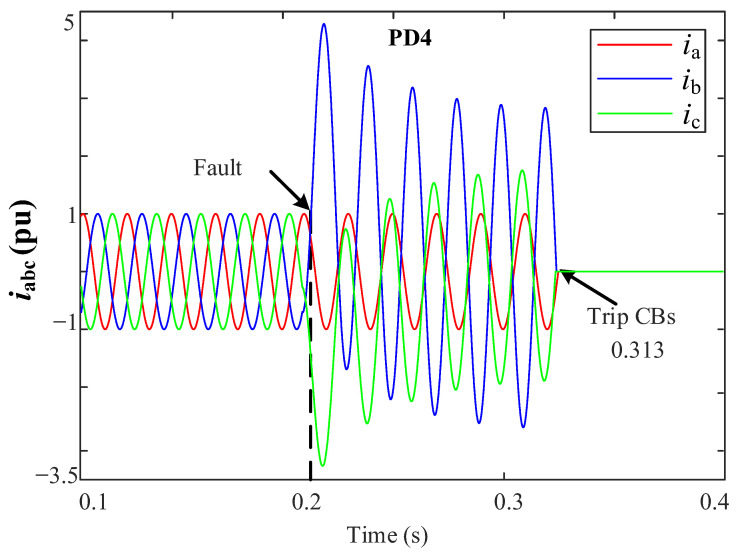
$i_{abc}$ currents for the DR during a 2PH fault at DL2, phases *b* and *c*.

**Figure 14 sensors-23-00980-f014:**
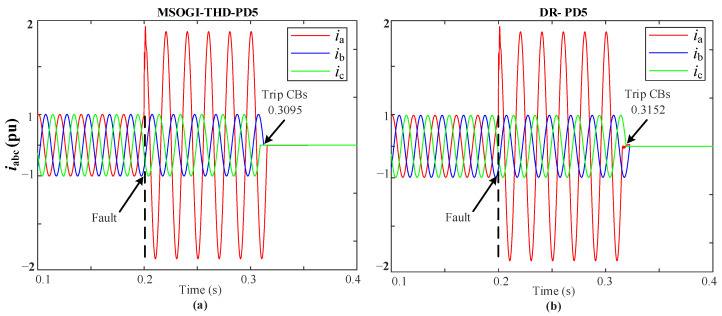
$i_{abc}$ currents during a 1PH-G (AG) fault at L3 using: (**a**) MSOGI-THD, (**b**) DR protection methods.

**Figure 15 sensors-23-00980-f015:**
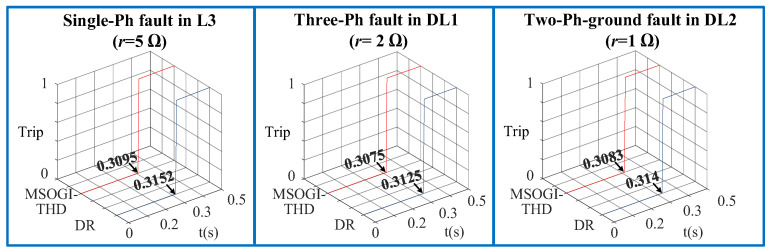
Trip signals of the MSOGI-THD and DR protection methods during symmetrical and unsymmetrical faults with different fault resistance.

**Table 1 sensors-23-00980-t001:** Comparison of common fault protection techniques for DS.

Protection Strategies	Advantages	Disadvantages	Communication Links	Cost
Directional Over Current Relay [11,12]	Have a good sensitivity for currents in both directions. An effective solution for bidirectional power flow problems.	Cannot be used in an all-contingencies scenario. Request the use of additional current Transformers (CTs) which implies an additional cost.	No	Expensive
Distance Relays [13,14]	Have a fast operation time. Unaffected by changes in the protected line.	Cannot distinguish between internal and external faults in the protected zone. The performance of the relay can be affected during power swing changes. Have an extra cost due to the required use of voltage transformers (VTs).	Yes	Reasonable
Differential Relays [15,16]	Have a high speed and sensitive fault detection response. Not affected by changes in the current’s flow direction and magnitude.	The implementation of the method is difficult.	Yes	Very Expensive
Voltage-based Protection [17,18]	Has a fast and sensitive fault identification response. Has a good ability for preventing a blackout. Can be used in different locations of the electrical network.	Impractical for complex networks. Any voltage drop within the network affects its performance. The relay settings need to be updated when network design is reconfigured.	Yes	Expensive
Adaptive Protection [19,20]	Relay settings are automatically readjusted to be compatible with the power system conditions.	Needs of advanced digital relays. Requires a high computational memory.	Yes	Expensive
Harmonic-Based Methods (FFT) [21]	Fast tripping response in comparison with other methods Can use the harmonic content of the grid’s voltage and current for the fault protection. No need for CTs or VTs.	The FFT supposes that a high computational burden will be implemented in a DSP, especially when it will be applied to each voltage line of the grid. Furthermore, the method cannot discriminate between phase-to-phase and phase-to-phase-to-ground faults.	Yes	Reasonable

**Table 2 sensors-23-00980-t002:** Parameters of the System.

Main Grid	MV Transformer (YNd11)	Distribution Lines						MV/LV Transformer (Dyn11)	DGs Rating	Load Rating	
		L3		DL1		DL2				Load1	Load2
Rated voltage 33 kV	Rated power 20 MVA	R	0.16 Ω/km	R	0.146 Ω/km	R	0.127 Ω/km	Rated power 400 kVA	320 kVA	480 kW	400 kW
		L	0.109 H/km	L	0.35 mH/km	L	25 mH/km				
	Rated voltage 33/11 kV	C	0.31 μF/km	C	0.31 μF/km	C	0.31 μF/km	Rated voltage 11/0.42 kV			
		Length 4 km		Length 3 km		Length 1 km					

**Table 3 sensors-23-00980-t003:** DR Settings.

Parameter	Value
Differential current (pu)	1.08
Biased characteristic (K)	0.5
Current transformer ratio (CT)	200:1

**Table 4 sensors-23-00980-t004:** Comparison of the proposed method with other methods operating under similar conditions.

References	Protection Strategies	Trip Time	Advantages	Disadvantages
[38]	OC and ANN	14 ms	Fast trip action, variable fault resistance	Communication problems, complex training process, not adaptable for network modifications.
[39]	Multi-Terminal DR	90 ms	Fast trip action, variable fault resistance	Communication problems.
[40]	Multi-Agent System and OCR	300 ms	No central controller	Communication problems.
[41]	Centralize Controller and Linear Programming	421 ms	No need for training, relay settings obtained simultaneously	Communication problems, more complex with large number of buses.
[42]	Over Current and Voltage Based	Not specified	Can locate the fault either inside the circuit breaker protection zone or not	Communication problems, undesirable for schemes with inverter-interfaced DG where the fault current flow is minimal.
[43]	THD	20–50 ms	Acts like a directional relay with no need of a voltage transformer..	Communication problems, validated only for three phase faults.
Proposed method	MSOGI-THD	7–10 ms	Fast tripping, variable fault locations, fault types, fault resistance, affordable computational burden.	Communication problems.
Proposed method minimal	MSOGI-III-THD	6.4 ms	The same merits of the proposed method before minimization. In addition, it is faster and a much affordable computational burden.	Communication problems.

**Table 5 sensors-23-00980-t005:** Fault detection and clearing times of the three MSOGI approaches with a 3PH-G fault at F3 and the number of cycles required to compute it.

Approach	Number of Cycles (c)	Fault Detection Time (ms)	Fault Clearing Time (s)
MSOGI	1788	7.0	0.3070
MSOGI-II	1341	6.6	0.3066
MSOGI-III	894	6.4	0.3064

## Data Availability

No new data were created in this study. Data sharing is not applicable to this article.

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
