# Peer review of "A THD-Based Fault Protection Method Using MSOGI-FLL Grid Voltage Estimator"

_sensors, 2023, doi:10.3390/s23020980_

Round 1
Reviewer 1 Report
This is a good quality work and can be published after minor changes with adding the comparison studies of some recent state of the art literature.
Reviewer 2 Report
This paper is well written, the paper structure is good, information is given in details. Just several minor issues.
1. It is better to add some quantitative results in the abstract to give a straight look of the algorithm performance.
2. I suggest to add a table for the comparison of conventional protection technologies, voltage based protection, adaptive protection and harmonic based methods, to show their key indicators, advantages, and disadvantages. This would give a clear sight to readers.
3. Although these protection methods are discussed individually, but a comprehensive comparison should be given, and also to show the current research gap, and why this study should be done. This should be done before mentioning what has been done in this paper.
4. Figure 3, in stage 3, all sub-routes should come to the end.
Reviewer 3 Report
Abstract:
overall fine - but correct grammar in Line 26:
"... a high accuracy and a faster response"
1. Introduction:
Line 31: wind energy systems. And insert a comma after this
Line 35: "Furthermore, several technical ...
Line 59: " and they have a high implementation cost"
Line 63: "However, this requires ... you need to spell out the acronym 'NN' once before using it - unless you mean 'ANN'
Line 64: remove "show to"
Line 85: "Harmonic-based methods ... [remove the "And, "]
Line 79: " ... since it needs advanced digital relays
Line 80: "requires a high computational ..."
Line 113: to the authors' ...
2. Protection Algorithm
Line 167, Table 1: Parameters of the System
Line 180: "Figure 2 shows ..." [drop in the 'In' at the beginning]
Line 195: the time in which
Line 199: optical fiber is characterized by its
Line 205: allowing it to stop, to wait ... and to return
3. Results and Discussion
Line 414: Comparison with different Protection Methods
Line 469: it was reported that ...
Line 483: Table 4
4. Conclusions
Line 506: maths: namely ...
References:
always:
In: Proccedings
capital P needed in [9] and [23]
[10] Renewable and Sustainable Energy Reviews
[41] IEEE Transactions on the Smart Grid
[44] IEEE Transactions on Industrial Applications
Reviewer 4 Report
The total harmonic distortion (THD) based fault protection method using MSOGI-FLL Grid voltage estimator is presented in the research. The manuscript is well written, however following are the suggestions to improve the quality of publication.
In abstract few lines can be added about the research problem along with the constraints in the existing solutions to highlight the novelty of the research.
Please place the Figure 2 and related table closer to the sub-section 2.1 proposed system currently it is placed under sub-section 2.2.
The states can be defined under the sub-section 2.3. finite state machine as State 1, State 2, 3, … State 6 without 2.3…. to avoid confusion.
Few lines about the future directions may be added at the end of concussions.
